# Measurement report: Molecular composition and volatility of gaseous organic compounds in a boreal forest: from volatile organic compounds to highly oxygenated organic molecules

Wei Huang[1,†,*], Haiyan Li[2,†], Nina Sarnela[1], Liine Heikkinen[1], Yee Jun Tham[1], Jyri Mikkilä[3], Steven J. Thomas[1], Neil M. Donahue[4], Markku Kulmala[1], and Federico Bianchi[1,*]

[1]Institute for Atmospheric and Earth System Research / Physics, Faculty of Science, University of Helsinki, Helsinki, 00014, Finland

[2]School of Civil and Environmental Engineering, Harbin Institute of Technology, Shenzhen, 518055, China

[3]Karsa Oy., A. I. Virtasen aukio 1, Helsinki, 00560, Finland

[4]Center for Atmospheric Particle Studies, Carnegie Mellon University, 5000 Forbes Avenue, Pittsburgh, PA 15213, USA

[†]*These authors contributed equally to this work.*

*Correspondence to*: Wei Huang (wei.huang@helsinki.fi) and Federico Bianchi (federico.bianchi@helsinki.fi)

**Abstract.** The molecular composition and volatility of gaseous organic compounds were investigated during April–July 2019 at the Station for Measuring Ecosystem – Atmosphere Relations (SMEAR) II situated in a boreal forest in Hyytiälä, southern Finland. In order to obtain a more complete picture and full understanding of the molecular composition and volatility of ambient gaseous organic compounds (from volatile organic compounds, VOCs, to highly oxygenated organic molecules, HOMs), two different instruments were used. A Vocus proton-transfer-reaction time-of-flight mass spectrometer (Vocus PTR-ToF; hereafter Vocus) was deployed to measure VOCs and less oxygenated VOCs (i.e., OVOCs). In addition, a multi-scheme chemical ionization inlet coupled to an atmospheric pressure interface time-of-flight mass spectrometer (MION API-ToF) was used to detect less oxygenated VOCs (using $Br^-$ as the reagent ion; hereafter MION-Br) and more oxygenated VOCs (including HOMs; using $NO_3^-$ as the reagent ion; hereafter MION-NO$_3$). The comparison among different measurement techniques revealed that the highest elemental oxygen-to-carbon ratios (O:C) of organic compounds were observed by the MION-NO$_3$ ($0.9 \pm 0.1$, average $\pm$ 1 standard deviation), followed by the MION-Br ($0.8 \pm 0.1$); and lowest by Vocus ($0.2 \pm 0.1$). Diurnal patterns of the measured organic compounds were found to vary among different measurement techniques, even for compounds with the same molecular formula, suggesting contributions of different isomers detected by the different techniques and/or fragmentation from different parent compounds inside the instruments. Based on the complementary molecular information obtained from Vocus, MION-Br, and MION-NO$_3$, a more complete picture of the bulk volatility of all measured organic compounds in this boreal forest was obtained. As expected, the VOC class was the most abundant (about 53.2 %), followed by intermediate-volatility organic compounds (IVOC, about 45.9 %). Although condensable organic compounds (low-volatility organic compounds, LVOC; extremely low-volatility organic compounds, ELVOC; and ultralow-volatility organic compounds, ULVOC) only comprised about 0.2 % of the total gaseous organic compounds, they play an important

role in new particle formation as shown in previous studies in this boreal forest. Our study shows the full characterization of the gaseous organic compounds in the boreal forest and the advantages of combining Vocus and MION API-ToF for measuring ambient organic compounds with different oxidation extent (from VOCs to HOMs). The results therefore provide a more comprehensive understanding of the molecular composition and volatility of atmospheric organic compounds as well as new insights in interpreting ambient measurements or testing/improving parameterizations in transport and climate models.

## 1 Introduction

Organic aerosol (OA) has significant impacts on climate (IPCC, 2013), air quality (Boers et al., 2015), and human health (Nel, 2005;Rückerl et al., 2011). Large amounts of biogenic and anthropogenic volatile organic compounds (VOCs) are emitted into the atmosphere (Atkinson and Arey, 2003), with biogenic VOCs (BVOCs) emissions greatly surpassing anthropogenic VOC emissions globally (Heald et al., 2008). The global BVOC emissions are dominated by terpenes (isoprene ($C_5H_8$), 594 Tg C $a^{-1}$; monoterpenes ($C_{10}H_{16}$), 95 Tg C $a^{-1}$; and sesquiterpenes ($C_{15}H_{24}$), 20 Tg C $a^{-1}$) (Sindelarova et al., 2014), which are mainly emitted by vegetation and can be influenced by meteorological conditions, such as temperature and light (Guenther et al., 1995;Kaser et al., 2013). After emission, they can undergo gas-phase oxidation with ozone ($O_3$), hydroxyl radical (OH), or nitrate radical ($NO_3$) forming thousands of oxygenated VOCs (i.e., OVOCs) with diverse functionalities that can be grouped into different volatility classes; Intermediate-volatility (IVOC), semi-volatile (SVOC), low-volatility (LVOC), extremely low-volatility (ELVOC), and ultralow-volatility (ULVOC). Organic compounds with sufficiently low volatility (e.g., LVOC, ELVOC, and ULVOC) can either form new particles or partition into the particle phase contributing to particulate growth and mass (Ehn et al., 2014;Bianchi et al., 2016;Bianchi et al., 2019;Simon et al., 2020;Schervish and Donahue, 2020;Kulmala et al., 2013). Recent studies have shown that highly oxygenated organic molecules (HOMs; Bianchi et al., 2019) are a major source of condensing or nucleating compounds and they play an important role in atmospheric new particle formation (Ehn et al., 2014;Bianchi et al., 2016;Kirkby et al., 2016;Tröstl et al., 2016;Bianchi et al., 2019;Kulmala et al., 1998). However, as a result of the complexity and analytical challenges of the precursor VOCs as well as the chemical composition and physicochemical properties of the resulting oxidation products (i.e., OVOCs), skill predicting their effects on air quality and climate is still limited.

Mass spectrometric techniques represent one general approach to investigate the chemical composition of organic compounds (Sullivan and Prather, 2005;Nash et al., 2006). One common ionization technique used in aerosol research is chemical ionization (CI; e.g., Caldwell et al., 1989;Ehn et al., 2014;Lopez-Hilfiker et al., 2014;Huang et al., 2019a). It is a soft ionization method (Gross, 2017) that utilizes the reactivity of the analyte towards the reagent ion to ionize molecules via transfer of an electron, proton, or other ions such as bromide and nitrate (Caldwell et al., 1989;Ehn et al., 2014;Sanchez et al., 2016;Yuan et al., 2017;Krechmer et al., 2018). Different chemical ionization mass spectrometers (CIMS) have different capabilities and sensitivities for detecting organic compounds (Riva et al., 2019). Proton-transfer-reaction mass spectrometry (PTR-MS) has been widely used to measure VOCs in the atmosphere (Yuan et al., 2017). The recently developed Vocus PTR time-of-flight mass spectrometer (Vocus PTR-ToF) has greatly enhanced sensitivity due to a newly designed chemical ionization source (Krechmer et al., 2018), and can detect a broader spectrum of VOCs (even diterpenes) and their oxygenated products (up to 6 to 8 oxygen atoms for monoterpene oxidation products; Li et al., 2020b). However, Vocus PTR-ToF is not preferred for detecting HOMs or dimers (Li et al., 2020b;Riva et al., 2019). The potential reason for the

latter case could be resulted from the fragmentation inside the instrument (Heinritzi et al., 2016) and/or losses in the sampling lines and on the walls of the inlet (Riva et al., 2019). The detection of less oxygenated VOCs (including less oxygenated dimers) and more oxygenated VOCs (including HOMs) can be well achieved by another instrument: an Atmospheric Pressure Interface Time-of-Flight mass spectrometer (API-ToF) coupled to a novel chemical ionization inlet, Multi-scheme chemical IONization inlet (MION; Rissanen et al., 2019). Via the fast switching between multiple reagent ion schemes (i.e., bromide and nitrate), it has been found that MION API-ToF is able to provide a more complete picture of the OVOCs for laboratory experiments performed in flow tube reactors (Rissanen et al., 2019). Br-CIMS has been found to have similar or even higher sensitivities than that of iodide-CIMS towards OVOCs depending on humidity (Hyttinen et al., 2018). It has also been used for the detection of hydroperoxyl radicals (Sanchez et al., 2016) and peroxy radicals formed by autoxidation (Rissanen et al., 2019). In addition to the molecular composition of organic compounds itself provided by the abovementioned state-of-the-art instruments (i.e., Vocus PTR-ToF and MION API-ToF), these information can also be used in volatility parameterizations to calculate effective saturation mass concentrations ($C_{sat}$) of individual organic compounds (Li et al., 2016;Donahue et al., 2011;Mohr et al., 2019), which can be then grouped into different volatility classes (or bins), i.e., volatility basis sets (VBS; e.g., Donahue et al., 2006;2011;2012;Cappa and Jimenez, 2010). However, due to the different instrumental capabilities and sensitivities as well as the lack of calibration standards for the majority of organic compounds for the different measurement techniques as abovementioned, it still remains challenging to provide a comprehensive understanding of the molecular composition and volatility of both VOCs and OVOCs, particularly in the field.

In the present work, we investigate the chemical composition and volatility of gaseous organic compounds (VOCs and OVOCs) measured during April and July 2019 in a boreal forest in Hyytiälä, southern Finland. The capabilities of the recently developed MION API-ToF for measuring ambient OVOCs are reported for the first time. Besides, the molecular composition and volatility of the OVOCs measured by MION API-ToF are compared and complemented with those as well as their precursor VOCs observed with Vocus PTR-ToF. With the combination of the organic compounds measured by both instruments, we present a more comprehensive picture of the molecular composition and volatility of the gaseous organic compounds in this boreal forest.

## 2 Methodology

### 2.1 Site description

The measurements were conducted between April 16–July 26, 2019 at the University of Helsinki Station for Measuring Ecosystem – Atmosphere Relations (SMEAR) II (Hari and Kulmala, 2005), which is located in a boreal forest in Hyytiälä, southern Finland (61°51′N, 24°17′E, 181 m a.s.l.). This station is dominated by Scots pine (*Pinus sylvestris*), and monoterpenes are found to be the dominating emitted biogenic non-methane VOCs (Barreira et al., 2017;Hakola et al., 2012). The measurement station has been considered as a rural background site (Manninen et al., 2010;Williams et al., 2011), and the nearest big city is Tampere, with more than 200,000 inhabitants and located ~60 km in the SW of our measurement site. A sawmill which is located 6–7 km away to the SE of our measurement site can contribute significantly to the OA loading in the case of SE winds, and the sawmill OA composition has been found to resemble biogenic OA a lot (Liao et al., 2011;Äijälä et al., 2017;Heikkinen et al., 2020).

**2.2 Measurements, quantification, and volatility calculation of gaseous organic compounds**

All mass spectrometers were set up in a temperature-controlled measurement container kept at ~25 °C. Sampling inlets were located about 1.5 m a.g.l. All data are reported in Eastern European Time (UTC+2).

**2.2.1 Measurements and quantification of gaseous organic compounds**

An API-ToF (Tofwerk Ltd.; equipped with a long ToF with a mass resolving power of ~9000) coupled to a recently developed multi-scheme chemical ionization inlet (MION, Karsa Ltd.; Rissanen et al., 2019) was used to analyze the molecular composition of OVOCs at a time resolution of 30 min. During the 30-min cycles of measurements, MION API-ToF switched modes among nitrate (NO$_3^-$, 8 min), bromide (Br$^-$, 8 min), and API (measuring natural ions, 10 min) modes, followed by 2 min of ion-filter zeroing for the API mode before switching from API mode to the next mode. More details about the instrument are well described by Rissanen et al. (2019). Gaseous organic compounds were sampled via a stainless steel tube (1 inch outer diameter) of ca. 0.9 m length and a flow rate of 20 L min$^{-1}$. Due to the large inlet diameter and flow rate, the SVOC and HOM losses are expected to be insignificant. Through the fast switching between the two reagent ion schemes, Br$^-$ and NO$_3^-$, less oxygenated VOCs (including less oxygenated dimers) and more oxygenated VOCs (including HOMs) can be measured, respectively (Rissanen et al., 2019). Data were analysed with the software packages, "tofTools" (developed by Junninen et al. (2010)) and "Labbis" (developed by Karsa Ltd.), which run in the MATLAB environment (MathWorks Inc., USA). Hereafter results from these two reagent ion schemes are abbreviated as MION-Br and MION-NO$_3$. The quantification of gaseous organic compounds measured with MION-Br and MION-NO$_3$ was calculated as in equation (1) and (2), respectively:

$$[\text{org}] = \frac{\text{org(Br}^-)}{\text{Br}^- + \text{H}_2\text{O(Br}^-)} \times C_{Br^-}, \tag{1}$$

$$[\text{org}] = \frac{\text{org(NO}_3^-)}{\sum_{i=0}^{2}(\text{HNO}_3^-)_i\,(\text{NO}_3^-)} \times C_{\text{NO}_3^-}. \tag{2}$$

where [org] is the concentration (unit: cm$^{-3}$) of the gaseous organic compound (obtained from high resolution fitting of each nominal mass) to be quantified; the numerators on the right-hand side are its detected signal clustered with bromide or nitrate, and the denominators are the sum of the reagent ion signals; $C_{Br^-}$ and $C_{\text{NO}_3^-}$ are the calibration factors representing the sensitivity of organic compound. The two stable isotopes of bromide ($^{79}$Br$^-$ and $^{81}$Br$^-$) share similar relative isotopic abundance, but only the compound clustered with $^{79}$Br$^-$ was used for the quantification (Sanchez et al., 2016), as the calibration factor, $C_{Br^-}$, was also calculated in a similar way. Following the approach by Rissanen et al. (2019), the calibration factors, $C_{Br^-}$ and $C_{\text{NO}_3^-}$, for sulphuric acid (H$_2$SO$_4$, compound representing the kinetic limit sensitivity; Viggiano et al., 1997; Berresheim et al., 2000) were determined to be $2.33 \times 10^{11}$ cm$^{-3}$ and $4.68 \times 10^{10}$ cm$^{-3}$, respectively. The calibration factors are higher than those reported by Rissanen et al. (2019) due to different instrumental settings and inlet setup. By comparing the ambient H$_2$SO$_4$ concentrations measured by MION-Br and MION-NO$_3$, the median value (0.53) was used to scale down the H$_2$SO$_4$ concentration measured by MION-Br, due to that the high water vapor concentrations in the calibration kit (~$5 \times 10^{14-16}$ cm$^{-3}$) might cause some uncertainties in the H$_2$SO$_4$ calibration factor of MION-Br (Hyttinen et al., 2018; Kürten et al., 2012). However, the MION-Br sensitivity has been found to be invariant with the measured ambient RH at our measurement site (20–100 %) for e.g. hydroperoxyl radicals (Sanchez et al., 2016), and the water clustered with Br$^-$ has also been included in the signal normalization of organic compounds to account for the humidity effect on reagent ion competition (see equation (1)). With the maximum sensitivity applied, the

concentrations therefore represent a lower limit. The uncertainties in the measured organic compound concentrations using calibration factors for $H_2SO_4$ have been reported to be ±50 % (Ehn et al., 2014) or a factor of 2 (Berndt et al., 2015). However, the uncertainties could be higher with variations in e.g. temperature and relative humidity (RH) in the field.

A Vocus PTR-ToF (Aerodyne Research Inc.; hereafter Vocus) was deployed to measure VOCs and less oxygenated VOCs at a time resolution of 5 s. During the measurements, the Vocus ionization source was operated at a pressure of 1.5 mbar. The ambient air was sampled via a polytetrafluoroethylene (PTFE) tube of ca. 1 m length and a total sample flow of 4.5 L min$^{-1}$. Of the total sample flow, around 100–150 cm$^3$ min$^{-1}$ went into the Vocus and the remainder was directed to the exhaust. The Vocus was automatically calibrated every three hours using a multi-component standard cylinder. The standard gases were diluted by the injection of zero air with a built-in active carbon filter, producing the VOCs mixing ratio of around 5 ppb. The sensitivity of VOCs measured by PTR instruments has been shown to relate to their elemental composition and functionality (Sekimoto et al., 2017). Some compounds were calibrated using authentic standards, including isoprene, monoterpenes, and some aromatic compounds. Compounds without authentic standards were divided into four different molecular groups, the CH (compounds with only carbon and hydrogen atoms), CHO (compounds with only carbon, hydrogen, and oxygen atoms), CHON (compounds with only carbon, hydrogen, oxygen, and nitrogen atoms), and others. Compounds with the formula of CH and CHO were quantified with the average sensitivities of the standards CH and CHO, respectively. For the groups of CHON and others, there was no standard available in the calibration mixture. We used the average sensitivity of all the CH and CHO standards to quantify CHON compounds and others. Quantification using the relationship between the kinetic reaction rate constants and calibrated sensitivity (Sekimoto et al., 2017; Yuan et al., 2017) did not show huge differences (slopes between 0.59–0.75; see Figure S1) for the concentrations of several CH species (e.g., sesquiterpenes and diterpenes) and several dominant CHO and CHON species (e.g., $C_7H_{10}O_4$, $C_8H_{12}O_4$, and $C_{10}H_{15}NO_{6-7}$), compared to the above-mentioned quantification method we used. The Vocus data analysis was performed using the software package "Tofware" (provided by Tofwerk Ltd.) that runs in the Igor Pro environment (WaveMetrics Inc., USA). Signals were pre-averaged over 30 min before the analysis.

When combining the organic compounds measured by the three different ionization techniques (i.e., MION-Br, MION-NO$_3$, and Vocus), for organic compounds observed in all ionization techniques the highest concentration was used. Background subtraction was performed for all spectra and therefore a lower signal for the same compound detected by any of the ionization techniques suggests a lower ionisation efficiency of the corresponding method (Stolzenburg et al., 2018).

**2.2.2 Volatility calculation of gaseous organic compounds**

Effective saturation mass concentrations ($C_{sat}$), a measure for volatility of a compound, were parameterized for each organic compound using the approach by Li et al. (2016) as in equation (3):

$$\log_{10} C_{sat} (298\ K) = (n_C^0 - n_C)b_C - n_O b_O - 2 \frac{n_C n_O}{(n_C + n_O)} b_{CO} - n_N b_N - n_S b_S \tag{3}$$

where $n_C$, $n_O$, $n_N$, and $n_S$ are the number of carbon, oxygen, nitrogen, and sulfur atoms in the organic compound, respectively; $n_C^0$ is the reference carbon number; $b_C$, $b_O$, $b_N$, and $b_S$ are the contribution of each atom to $\log_{10} C_{sat}$, respectively; $b_{CO}$ is the carbon–oxygen nonideality (Donahue et al., 2011). These "$b$" values depend on the composition of precursor gases, depending largely on whether the precursors are aliphatic (including terpenes) or

aromatic. In addition to being derived from literature structure activity relations (i.e., SIMPOL; Pankow and Asher, 2008), the relations have been quantitatively confirmed for both aliphatic and aromatic systems using filter inlet for gases and aerosols (FIGAERO) thermal desorption CIMS measurements on carefully controlled precursor oxidation experiments at the CLOUD (Cosmics Leaving Outdoor Droplets) facility at CERN (European Organization for Nuclear Research) (Ye et al., 2019;Wang et al., 2020). For the boreal forest conditions in this work we use the aliphatic (more volatile) parameterization and these "*b*" values can be found in Li et al. (2016). Due to that the empirical approach by Li et al. (2016) was derived with very few organonitrates and could therefore lead to bias for the estimated vapor pressure (Isaacman-VanWertz and Aumont, 2020), we modified the $C_{sat}$ (298 K) of CHON compounds by replacing all $NO_3$ groups as OH groups (Daumit et al., 2013).

To obtain the $C_{sat}$ (T), we adjusted the $C_{sat}$ (298 K) (Donahue et al., 2011;Epstein et al., 2010) to the measured ambient temperature as in equations (4) and (5):

$$\log_{10} C_{sat}(T) = \log_{10} C_{sat}(298\text{ K}) + \frac{\Delta H_{vap}}{R\ln(10)}\left(\frac{1}{298} - \frac{1}{T}\right), \tag{4}$$

$$\Delta H_{vap}(\text{kJ mol}^{-1}) = -11 \cdot \log_{10} C_{sat}(298\text{ K}) + 129. \tag{5}$$

where $T$ is the temperature in Kelvin; $C_{sat}$ (298 K) is the saturation mass concentrations at 298 K; $\Delta H_{vap}$ is the vaporization enthalpy; $R$ is the gas constant (8.3143 J K$^{-1}$ mol$^{-1}$).

Uncertainties arising from the potential presence of isomers is limited within this dataset, since they cannot be differentiated using the formula-based parameterization with the only input being the molecular composition. Accuracy to within 1 order of magnitude for terpene oxidation products has been confirmed by calibrated thermal desorption measurement (Wang et al., 2020) and by closure with size-resolved growth rate measurements at the CLOUD experiment (Stolzenburg et al., 2018). Besides, the fragmentation of organic compounds inside the instruments (e.g., Vocus) may also bias the $C_{sat}$ results towards higher volatilities, resulted from the signal bias of parent ions towards lower values and of fragment ions towards higher values (Heinritzi et al., 2016).

## 2.3 Additional co-located measurements

The meteorological parameters were continuously monitored at this measurement site. Temperature was monitored with a Pt100 sensor (Platinum resistance thermometer with a resistance of 100 ohms (Ω) at 0 °C) inside ventilated custom-made radiation shield, while wind directions and wind speed with a 2D Ultrasonic anemometer (Adolf Thies GmbH & Co. KG), and the global radiation with an EQ08 pyranometer (Carter-Scott Manufacturing Pty. Ltd.). The main wind direction above the canopy during the measurement period was southwest (see Figure S2). The mixing ratios of ozone ($O_3$) and nitrogen oxides (NO and $NO_2$) were measured with an ultraviolet light absorption analyzer (TEI 49C, Thermo Fisher Scientific Inc.) and a chemiluminescence analyzer (TEI 42CTL, Thermo Fisher Scientific Inc.), respectively. The mixing ratios of sulfur dioxide ($SO_2$) were measured with a fluorescence analyzer (TEI 43CTL, Thermo Fisher Scientific Inc.).

An aerosol chemical speciation monitor (ACSM; Aerodyne Research Inc.; Ng et al., 2011) was deployed to continuously measure the non-refractory sub-micrometer aerosol particle chemical composition. The ACSM, which contains a quadrupole mass spectrometer, provided unit-mass resolution mass spectra every 30 min. This information was chemically speciated to organic, sulfate, nitrate, ammonium, and chloride concentrations by the ACSM analysis software. The mass concentrations of each species were calculated based on frequently conducted ionization efficiency calibrations. The data were corrected for collection efficiency, which was ca. 60 % during the measurement period. The sampling was conducted through a PM$_{2.5}$ cyclone and a Nafion dryer (RH < 30 %)

with a stainless steel tube of ca. 3 m length and a flow rate of 3 L min$^{-1}$ (only 1.4 cm$^3$ s$^{-1}$ into the ACSM). The recorded data were analyzed using the ACSM local v. 1.6.0.3 toolkit (provided by Aerodyne Research Inc.) within the Igor Pro v. 6.37 (Wavemetrics Inc., USA). More details about ACSM operation and data processing can be found in Heikkinen et al. (2020).

## 3 Results and discussion

### 3.1 Overview of the measurements

Figure 1 shows the overview of the time series of meteorological parameters (temperature, global radiation, and wind direction and wind speed), trace gas concentrations (SO$_2$, O$_3$, NO, and NO$_2$), and total gaseous organic compounds measured by MION-Br, MION-NO$_3$, and Vocus, as well as total particulate organics measured by ACSM. Note that relatively long-lived compounds like ethanol, acetone, and acetic acid, are excluded from Vocus data presented in this study in order to focus on compounds actively involved in the fast photochemistry (all excluded compounds are listed in Table S1 and the time series of total organic compound concentrations including them are shown in Figure S3). As we can see from Figure 1a, most of the measurement days had strong photochemical activity with ambient temperature exhibiting clear diurnal patterns ranging between –3 and 32 °C. In general, the time series of the total organics (both gas phase and particle phase; see Fig. 1e–f) measured by MION-Br, MION-NO$_3$, Vocus, and ACSM were similar during the measurement period. Elevated levels of total gaseous and particulate organics (e.g., May 17–24 and June 7–10; see Fig. 1e–f) were observed at warmer days with strong global radiation and the main wind direction coming from southeast (the direction of the sawmill; for e.g. May 17–24) or southwest (for e.g. June 7–10; see Fig. 1a–b). Besides, higher concentrations of oxidants of VOCs (such as O$_3$) and/or anthropogenic pollutants (such as SO$_2$ and NO$_x$) also followed some of the elevated concentrations of gaseous and/or particulate organics (e.g., April 19–May 3, May 17–24, and June 7–10; see Fig. 1c–d). The observations of the elevated organics could be resulted from higher VOC emissions (e.g., terpenes, the typically observed VOCs, Li et al., 2020a; Figure S3) influenced by meteorological conditions (i.e., temperature and/or light; Guenther et al., 1995;Kaser et al., 2013), different air mass origins (e.g., terpene pollutions from the sawmill in the case of SE winds; Liao et al., 2011;Äijälä et al., 2017;Heikkinen et al., 2020), as well as chemistry initiated by/related with different trace gases (Yan et al., 2016;Massoli et al., 2018;Huang et al., 2019b;Heikkinen et al., 2020). The results suggest the important roles meteorological parameters, trace gases, and air masses play in the emission and oxidation reactions of organic compounds. Due to the soft ionization processes of organic molecules in the Vocus, MION-Br, and MION-NO$_3$, molecular composition of organic compounds was obtained. In the next section we will discuss the molecular composition of gaseous organic compounds measured by Vocus, MION-Br, and MION-NO$_3$.

### 3.2 Molecular composition of gaseous organic compounds

During the measurement period, Vocus identified 72 CH compounds (C$_{x\geq1}$H$_{y\geq1}$) and 431 CHOX compounds (C$_{x\geq1}$H$_{y\geq1}$O$_{z\geq1}$X$_{0-n}$), with X being different atoms like N, S, or a combination thereof, while MION-Br and MION-NO$_3$ detected 567 and 687 CHOX compounds, respectively. Substantial overlaps of organic compounds were observed for these three ionization techniques while distinct organic compounds were also detected with individual method (Figure S4). The average mass-weighted chemical compositions for organic compounds measured by

Vocus, MION-Br, and MION-NO$_3$ were C$_{5.3}$H$_{7.5}$O$_{1.1}$N$_{0.1}$, C$_{6.7}$H$_{10.7}$O$_{4.3}$N$_{0.3}$, and C$_{7.5}$H$_{11.4}$O$_{5.4}$N$_{0.3}$, respectively. We stress here that the fragmentation of organic compounds inside the Vocus may bias the chemical composition towards shorter carbon backbone. And the average mass-weighted chemical composition representing the bulk of all measured gaseous organic compounds (with the approach described in section 2.2.1) in this boreal forest was calculated to be C$_{6.0}$H$_{8.7}$O$_{1.2}$N$_{0.1}$, indicative of the short carbon backbone and relatively low oxidation extent.

Similar to previous laboratory results (Riva et al., 2019), MION-NO$_3$ observed the most oxidized compounds with the highest elemental oxygen-to-carbon ratios (O:C; 0.9 ± 0.1, average ± 1 standard deviation), followed by the MION-Br (0.8 ± 0.1); the O:C of the organics detected by Vocus were lowest (0.2 ± 0.1). In addition, CHO group comprised the largest fraction of the total organic compounds (Vocus: 43.6 ± 9.4 %; MION-Br: 75.4 ± 5.3 %; MION-NO$_3$: 71.8 ± 7.9 %; see Table 1). The second most abundant group for Vocus was CH group making up 35.2 ± 15.1 % of its total organic compounds; while it was CHON group for MION-Br (24.1 ± 5.2 %) and MION-NO$_3$ (28.1 ± 7.9 %; see Table 1), indicating active NO$_x$ or NO$_3$ radical related chemistry (Yan et al., 2016). CHON group only accounted for 8.1 ± 2.7 % of the total organic compounds measured by Vocus, possibly due to its lower sensitivity towards larger organonitrates (see also Fig. S5) caused by their losses in the sampling lines and on the walls of the inlet (Riva et al., 2019) and/or fragmentation inside the instrument (Heinritzi et al., 2016).

The mass defect plots for organic compounds measured by Vocus, MION-Br, and MION-NO$_3$ are shown in Figure 2. Similar to previous studies (e.g., Yan et al., 2016;Li et al., 2020a), multiple series of organic compounds with different number of carbon atoms (such as C$_5$, C$_{10}$, C$_{15}$, and C$_{20}$) and oxygen atoms (up to 20; see also Fig. S5) were measured in this boreal forest environment. Organics with the lowest oxidation extent were better observed by Vocus, while organics with the largest molecular weights and highest oxidation extent were better observed by MION-NO$_3$ (Fig. 2a). Figure 2b shows the mass defect plots of organic compounds grouped into different categories. The markers are color-coded with different compound groups, such as CH, CHO, CHON, and others. The size of the markers is proportional to the logarithm of the concentration of each compound. Generally, similar to previous laboratory results (Riva et al., 2019;Rissanen et al., 2019), Vocus and MION-Br detected better the CHO compounds in the mass range of 50–100 Da and CHON compounds in the mass range of 50–150 Da, and MION-Br even CHON compounds in the mass range of 350–425 Da, which are most likely to be less oxygenated monomers or dimers; while MION-NO$_3$ was more sensitive towards the CHO and CHON compounds in the mass range of 425–600 Da, which are most likely to be more oxygenated HOM dimers (see Fig. 2b and Fig. S5).

We further investigated the contributions of the measured CHOX compounds with different number of oxygen atoms per molecule to total CHOX compounds as a function of the number of carbon atoms (Figure 3). Organic compounds which were detected with higher sensitivity by Vocus were those with the number of carbon atoms between 3 and 10 and the number of oxygen atoms between 1 and 3 (i.e., less oxygenated monomers); compounds with larger number of carbon atoms (i.e., >10) and oxygen atoms (i.e., >3) were much better detected by MION-Br and MION-NO$_3$; the former particularly for CHON compounds with the number of carbon atoms between 15 and 20 and oxygen atoms between 4 and 8 (i.e., larger less oxygenated monomers and dimers; see Fig. S5b) and the latter particularly for compounds with the number of oxygen atoms larger than 9 (i.e., HOM monomers and dimers; Rissanen et al., 2019;Riva et al., 2019;Li et al., 2020b; see Fig. 3 and Fig. S5). In the MION-Br and MION-NO$_3$ data, CHOX compounds with the number of carbon atoms of 5, 10, 15, and even 20 exhibited relatively elevated contributions compared to their neighbours (Fig. 3), indicating contributions of their potential corresponding precursors, i.e., isoprene, monoterpenes, sesquiterpenes, and diterpenes (together accounting for 38.3 ± 12.5 % of total CH compounds; see Table S2, Fig. S3, and Fig. S6). We emphasize here that using the

number of carbon atoms as a basis to relate the CHOX to their precursor VOCs is a simplified assumption, as negative or positive artifacts can arise from fragmentation or accretion reactions (Lee et al., 2016). Similar pattern was also observed by Huang et al. (2019a) in a rural area in southwest Germany, based on filter inlet for gases and aerosols high-resolution time-of-flight chemical ionization mass spectrometer (FIGAERO-HR-ToF-CIMS) data. The consistency and complement of the results demonstrate the different capabilities of these instruments for measuring gaseous organic compounds with different oxidation extent (from VOCs to HOMs).

### 3.3 Diurnal characteristics of gaseous organic compounds

Median diurnal variations of total CH, total CHO, and total CHON compounds measured by Vocus, MION-Br, and MION-$NO_3$ are shown in Figure 4. In general, the CH and CHO group measured by Vocus exhibited higher levels during the night (see Fig. 4a–b), mainly driven by the boundary layer height dynamics (Baumbach and Vogt, 2003;Zha et al., 2018). Besides, CHO compounds measured by Vocus were dominated by $O_{1-2}$ compounds (see Fig. 3 and Fig. S5) and have also been reported to follow more the CH trends (Li et al., 2020b). Their relatively flat diurnal pattern could be resulted from the smearing effect after summing up the much less oxygenated CHO molecules (mostly peak at night) and comparatively more oxygenated CHO molecules (mostly peak during daytime) (Li et al., 2020b). In contrast, the CHO and CHON group measured by MION-Br and MION-$NO_3$ exhibited higher levels during the day (see Fig. 4b), due to strong photochemical oxidation caused by different meteorological parameters (i.e., temperature and global radiation; see Fig. 1a and Fig. S7) and/or elevated trace gas levels (e.g., $O_3$ and $SO_2$; see Fig. 1c and Fig. S7; Yan et al., 2016;Massoli et al., 2018;Huang et al., 2019b;Bianchi et al., 2017). However, the CHON group measured by Vocus showed relatively stable signals throughout the day (see Fig. 4c). The potential reason could be partly due to its lower sensitivity towards larger organonitrates (see Fig. S5) caused by their losses in the sampling lines and on the walls of the inlet (Riva et al., 2019) and/or their fragmentation inside the instrument (Heinritzi et al., 2016). Another potential reason could be resulted from the smearing effect after summing up the much less oxygenated CHON molecules (mostly peak at night or early morning) and comparatively more oxygenated CHON molecules (mostly peak during daytime) (Li et al., 2020b).

Different diurnal patterns among different measurement techniques can also be found for individual organic compounds with the same molecular formula, such as several dominant CHO and CHON species, $C_7H_{10}O_4$ (molecular formula corresponding to 3,6-oxoheptanoic acid identified in the laboratory as limonene oxidation product by Faxon et al., 2018;Hammes et al., 2019), $C_8H_{12}O_4$ (molecular formula corresponding to terpenylic acid identified in monoterpene oxidation product by Zhang et al., 2015;Hammes et al., 2019), and $C_{10}H_{15}NO_{6-7}$ (identified in the laboratory as monoterpene oxidation products by Boyd et al., 2015;Faxon et al., 2018; see Figure 5). The inconsistent trends in time series and the varying correlations of these above-mentioned dominant CHO and CHON species indicate different isomer contributions detected by different measurement techniques (Figure S8 and Table S3). Similar behaviors were also evident for OVOCs with varying oxidation extent, like the terpene-related $C_xHO$ and $C_xHON$ compounds (x = 5, 10, 15, and 20; see Figure. S9), which in total accounted for up to 27 % and 39 % of their corresponding CHO and CHON groups (see Table S2). Most of the terpene-related $C_xHO(N)$ groups (x = 5, 10, 15, and 20) with different oxidation extent behaved similar among different measurement techniques, but some were also found to vary (see Fig. S9). Compounds with the same number of carbon and oxygen atoms but different hydrogen atoms (i.e., different saturation level) were also found to behave differently (see Fig. S9c–d), possibly due to different chemistry involved in their formation (Zhao et al., 2018; Molteni et al.,

2019). Even compounds with the same molecular formula varied among different measurement techniques (see Fig. S9c–d and also Fig. 5). The differences can be likely resulted from different isomers detected by the different techniques, and/or fragmentation products from different parent compounds inside the instruments (e.g., Heinritzi et al., 2016;Zhang et al., 2017).

The results indicate that organic compounds may behave differently among different measurement techniques during different time period. In the next section, we will investigate the volatility of these gaseous organic compounds, which can influence their lifetime and roles in the atmosphere.

### 3.4 Volatility of organic compounds

Based on the $\log_{10}C_{sat}$ values of all organic compounds parameterized with the modified Li et al. (2016) approach (Daumit et al., 2013;Isaacman-VanWertz and Aumont, 2020) described in section 2.2.2, they were grouped into a 25-bin volatility basis set (VBS; Donahue et al., 2006) (Figure 6a). Organic compounds with $C_{sat}$ lower than $10^{-8.5}$ µg m$^{-3}$, between $10^{-8.5}$ and $10^{-4.5}$ µg m$^{-3}$, between $10^{-4.5}$ and $10^{-0.5}$ µg m$^{-3}$, between $10^{-0.5}$ and $10^{2.5}$ µg m$^{-3}$, between $10^{2.5}$ and $10^{6.5}$ µg m$^{-3}$, and higher than $10^{6.5}$ µg m$^{-3}$ are termed ULVOC, ELVOC, LVOC, SVOC, IVOC, and VOC, respectively (Donahue et al., 2009;Schervish and Donahue, 2020). The VBS resulting pie charts for these compound groups and their mean contributions are shown in Figure 6b–d and Table 2. Organic compounds with $C_{sat}$ of $10^4$ µg m$^{-3}$ made up the biggest mass contributions for MION-Br and MION-NO$_3$, and the dominating $C_{sat}$ bin measured by Vocus was organic compounds with $C_{sat}$ of $10^6$ µg m$^{-3}$ (see Fig. 6a). Furthermore, Vocus observed much higher contributions of VOC with $C_{sat}$ higher than $10^8$ µg m$^{-3}$, whereas MION-NO$_3$ higher contributions of ELVOC and ULVOC with $C_{sat}$ lower than $10^{-8}$ µg m$^{-3}$ (See Fig. 6a). And MION-Br and MION-NO$_3$ observed comparable contributions of compounds with $C_{sat}$ between $10^{-7}$ and $10^7$ µg m$^{-3}$. We stress here that the fragmentation of organic compounds inside the Vocus may bias the $C_{sat}$ results towards higher volatilities.

IVOC, which include generally less oxygenated VOCs, comprised the significant fraction of total organics (Vocus: $45.8 \pm 5.4$ %; MION-Br: $65.8 \pm 8.5$ %; MION-NO$_3$: $56.3 \pm 10.6$ %), indicating substantial oxidation extent of the precursor VOC, which made up $53.7 \pm 5.5$ % of the total organics measured by Vocus but much less by MION-Br ($10.4 \pm 8.2$ %) and MION-NO$_3$ ($5.4 \pm 2.4$ %; see Fig. 6b–d and Table 2). SVOC, which include slightly more oxygenated VOCs, constituted substantially (Vocus: $0.4 \pm 0.2$ %; MION-Br: $16.2 \pm 4.9$ %; MION-NO$_3$: $23.9 \pm 5.1$ %) to the measured organic compounds. LVOC and ELVOC, which include OVOCs with higher oxidation degrees and mainly contribute to the growth of embryonic clusters in the atmosphere (Donahue et al., 2012;Bianchi et al., 2019), accounted for >8 % of the corresponding total organics measured by MION-Br and MION-NO$_3$; while ULVOC, which include OVOCs with even higher oxidation extent that are the most effective drivers of pure biogenic nucleation (Schervish and Donahue, 2020;Simon et al., 2020), accounted for $0.5 \pm 0.6$ % of total organics measured by MION-NO$_3$ (see Fig. 6b–d and Table 2). Differences in the contribution of these compound groups (Fig. 6b–d and Table 2) could be due to different sensitivities of the instruments towards organic compounds with varying oxidation extent (Riva et al., 2019).

With the complementary molecular information of organic compounds from Vocus, MION-Br, and MION-NO$_3$, a combined volatility distribution was plotted to estimate the bulk volatility of all measured organic compounds (with the approach described in section 2.2.1) at our measurement site (Figure 7). The combined volatility distribution covers very well from VOCs to HOMs, with varying O:C ratios and volatility ranges (Figure 7a). It therefore provides a more complete picture of the volatility distribution of gaseous organic compounds in

this boreal forest. The average mass-weighted $\log_{10}C_{sat}$ value representing the bulk of all measured gaseous organic compounds in this boreal forest was ~6.1 µg m$^{-3}$. In general, MION-NO$_3$ measured >91 % of the ULVOC while MION-Br measured >70 % of the ELVOC, and Vocus >98 % of the IVOC, and VOC (Figure S10). As we can see from Fig. 7b, the VOC class was found to be the most abundant (about 53.2 %), followed by the IVOC (about 45.9 %), indicating that the bulk gaseous organic compounds observed in this boreal forest were relatively fresh,

which is also consistent with the bulk molecular composition's relatively low oxidation extent. Differences of the bulk volatility of organic compounds between daytime (between 10:00 and 17:00) and nighttime (between 22:00 and 05:00) were not significant (Figure S11). Given the location of the measurement station that is inside a boreal-forested area, the gaseous organic compounds were expected to be dominated by VOC and IVOC. The abundance of the CH compounds such as terpenes (see Table 1, Table S2, Fig. S3, and Fig. S6) as well as less oxygenated

VOC (see Fig. 3 and Fig. S5) support this conclusion. Although the condensable vapors (LVOC, ELVOC, and ULVOC) only comprised about 0.2 % of the total gaseous organic compounds, they contribute significantly to forming new particles via nucleation and further particulate growth and mass via condensation in this boreal forest (Kulmala et al., 2013;Ehn et al., 2014;Mohr et al., 2019). The results from the combined VBS could provide a better basis to test and improve parameterizations for predicting organic compound evolutions in transport and

climate models.

**4 Conclusions**

In this paper, with an aim of obtaining a more complete picture from VOCs to HOMs, the molecular composition and volatility of gaseous organic compounds were investigated with the deployment of a Vocus and a MION API-ToF during April–July 2019 at the SMEAR II station situated in a boreal forest in Hyytiälä, southern Finland.

Similar to previous laboratory results (Riva et al., 2019), highest elemental O:C ratios of organic compounds were observed by the MION-NO$_3$ (0.9 ± 0.1), followed by the MION-Br (0.8 ± 0.1), and lowest by the Vocus (0.2 ± 0.1). Different from the pattern observed by Vocus which were mostly dominated by compounds with the number of carbon atoms between 3 and 10 and the number of oxygen atoms between 1 and 3 (i.e., less oxygenated monomers), compounds with larger number of carbon atoms (i.e., >10) and oxygen atoms (i.e., >3) were much

better detected by MION-Br (particularly for larger less oxygenated monomers and dimers) and MION-NO$_3$ (particularly for HOM monomers and dimers). The average mass-weighted chemical composition representing the bulk of all measured gaseous organic compounds in this boreal forest was $C_{6.0}H_{8.7}O_{1.2}N_{0.1}$, indicative of the short carbon backbone and relatively low oxidation extent. Besides, diurnal patterns of the measured organic compounds were found to vary among different measurement techniques, even for compounds with the same molecular

formula. The results indicate contributions of different isomers detected by the different techniques and/or fragmentation products from different parent compounds inside the instruments (e.g., Heinritzi et al., 2016;Zhang et al., 2017).

From the more complete picture of the bulk volatility of all measured organic compounds in this boreal forest, the average mass-weighted $\log_{10}C_{sat}$ value representing the bulk of all measured gaseous organic compounds in

this boreal forest was ~6.1 µg m$^{-3}$. In addition, the VOC class was found to be the most abundant (about 53.2 %), followed by the IVOC (about 45.9 %), indicating that the bulk gaseous organic compounds were relatively fresh, consistent with the bulk molecular composition's relatively low oxidation extent. Although condensable organic compounds (LVOC, ELVOC, and ULVOC) only comprised about 0.2 % of the total gaseous organic compounds,

they play an important role, forming new particles via nucleation and contributing to particulate growth and mass

via condensation in this boreal forest (Kulmala et al., 2013;Ehn et al., 2014;Mohr et al., 2019).

The results show the full characterization of the gaseous organic compounds in the boreal forest, and the advantages of combining Vocus and MION API-ToF for measuring ambient gaseous organic compounds with different oxidation extent (from VOCs to HOMs). Our study provides a more comprehensive understanding of the molecular composition and volatility of atmospheric organic compounds, as well as new insights when interpreting

ambient measurements or using them as input to test and improve parameterizations for predicting organic compound evolutions in transport and climate models.

**Data availability**

Data are available upon request to the corresponding authors.

**Author contributions**

WH analyzed the MION API-ToF data, produced all figures, and wrote and edited the paper; HL operated and calibrated Vocus, analysed the Vocus data, provided suggestions for the data analysis, interpretation and discussion, and edited the paper; NS operated and calibrated MION API-ToF, preprocessed the MION API-ToF data, and provided suggestions for the data analysis, interpretation, and discussion; LH performed ACSM measurements, analyzed the ACSM data, and provided suggestions for the data interpretation and discussion; YJT

provided suggestions for the data interpretation and discussion; JM helped with the MION measurements and provided suggestions for the data interpretation and discussion; SJT helped with the Vocus measurements; NMD provided suggestions for the data interpretation and discussion; MK organized the campaign and provided suggestions for the data interpretation and discussion; FB organized the campaign, provided suggestions for the data analysis, interpretation, and discussion, and edited the paper. All authors contributed to the final text.

**Competing interests**

The authors declare no conflict of interest.

**Acknowledgements**

This work was supported by the staff at INAR. Hyytiälä personnel is acknowledged for their help in conducting the measurements. J. Hakala is acknowledged for his help with MION measurements. J. Ma is acknowledged for

his technical help with data analysis. We thank the tofTools team and Karsa labbis team for providing tools for mass spectrometry data analysis. This work was supported by the Academy of Finland (Nr. 311932), the European Research Council with the grant CHAPAs (Nr. 850614), European Research Council via ATM-GTP 266 (742206), Jane and Aatos Erkko Foundation, and US NSF grant AGS1801897.

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

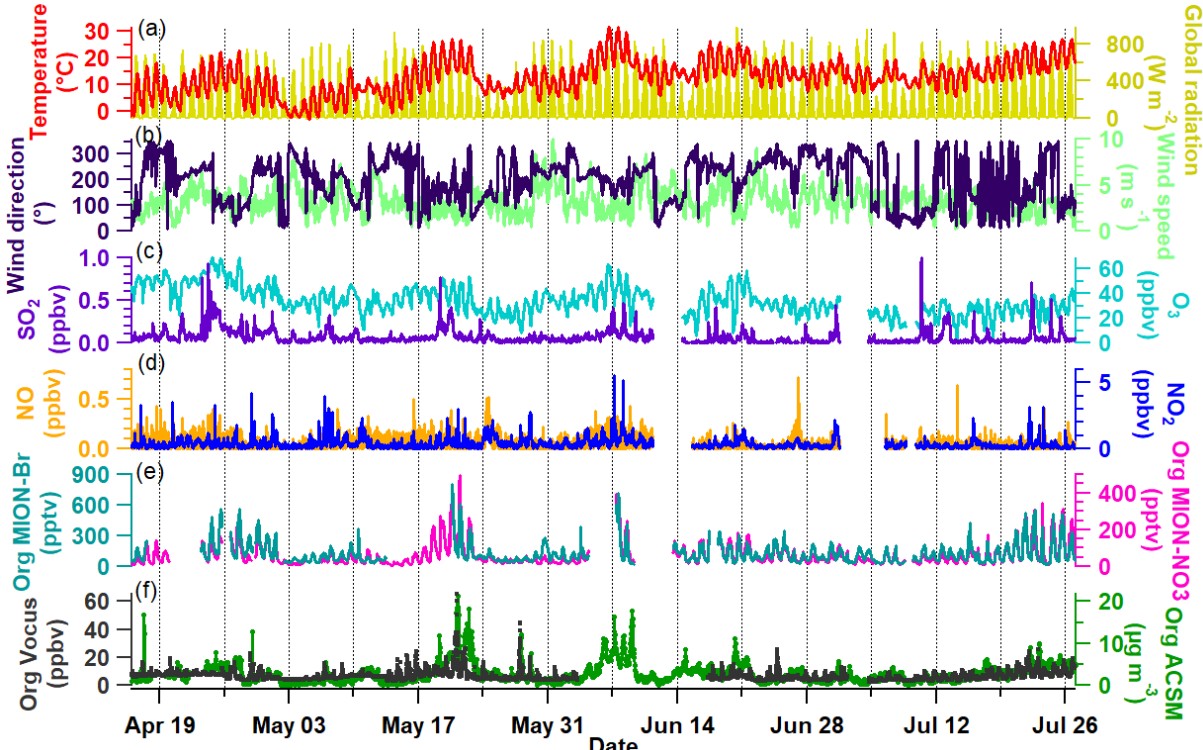

**Figure 1.** Overview of the time series from April 16 to July 26, 2019. **(a)** temperature and global radiation; **(b)** wind direction and wind speed; **(c)** mixing ratios of $SO_2$ and $O_3$; **(d)** mixing ratios of NO and $NO_2$; **(e)** total gaseous organics measured by MION-Br and MION-$NO_3$; and **(f)** total gaseous organics measured by Vocus as well as total particulate organics measured by ACSM. The data gap between MION-Br and MION-$NO_3$ (e.g., around May 17) was due to that the MION API-ToF was only running with API mode and $NO_3$ mode because of a mass flow controller issue for Br mode at that time.

765    **Table 1.** Contribution (%, average $\pm$ 1 standard deviation) of different compound groups to total organics measured by different measurement techniques.

| Compound group | Vocus | MION-Br | MION-NO$_3$ |
|---|---|---|---|
| CH | 35.2 $\pm$ 15.1 % | - | - |
| CHO | 43.6 $\pm$ 9.4 % | 75.4 $\pm$ 5.3 % | 71.8 $\pm$ 7.9 % |
| CHON | 8.1 $\pm$ 2.7 % | 24.1 $\pm$ 5.2 % | 28.1 $\pm$ 7.9 % |
| others | 13.1 $\pm$ 3.9 % | 0.5 $\pm$ 0.6 % | 0.1 $\pm$ 0.1 % |

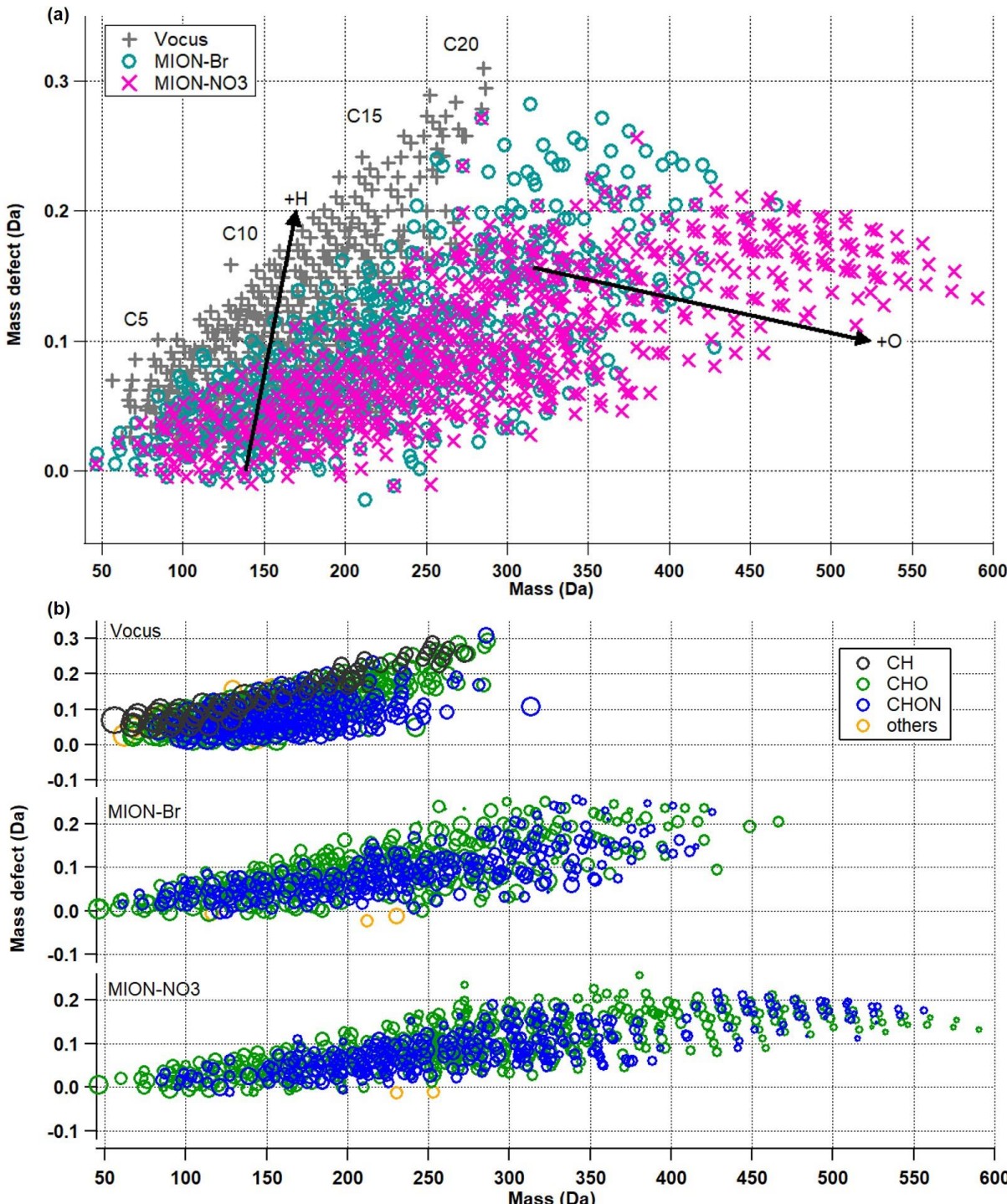

**Figure 2.** **(a)** Mass defect plots for organic compounds measured by Vocus, MION-Br, and MION-NO₃; **(b)** mass defect plots for organic compounds (separated into CH, CHO, CHON, and others) measured by Vocus, MION-Br, and MION-NO₃. Markers in **(b)** were all sized by the logarithm of their corresponding concentrations.

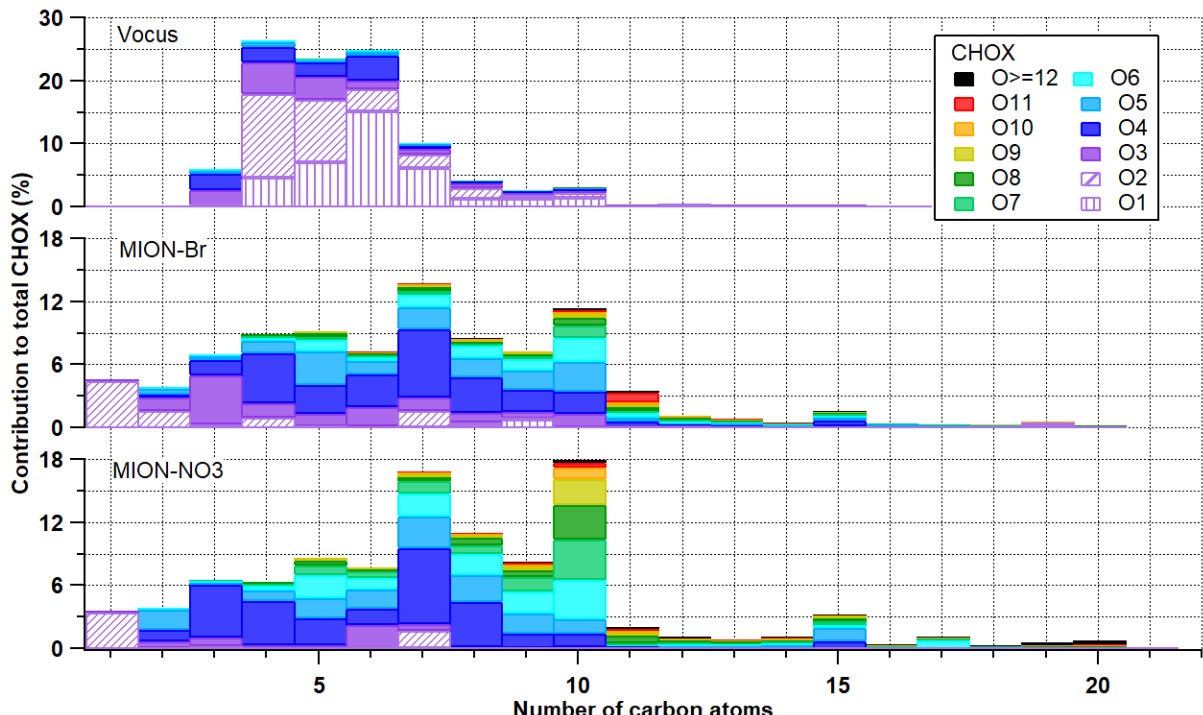

**Figure 3.** Contribution of measured CHOX compounds with different number of oxygen atoms to total CHOX compounds as a function of the number of carbon atoms for Vocus (upper panel), MION-Br (middle panel), and MION-NO$_3$ (bottom panel). Vocus panel has excluded CHX compounds (i.e., O$_0$ compounds).

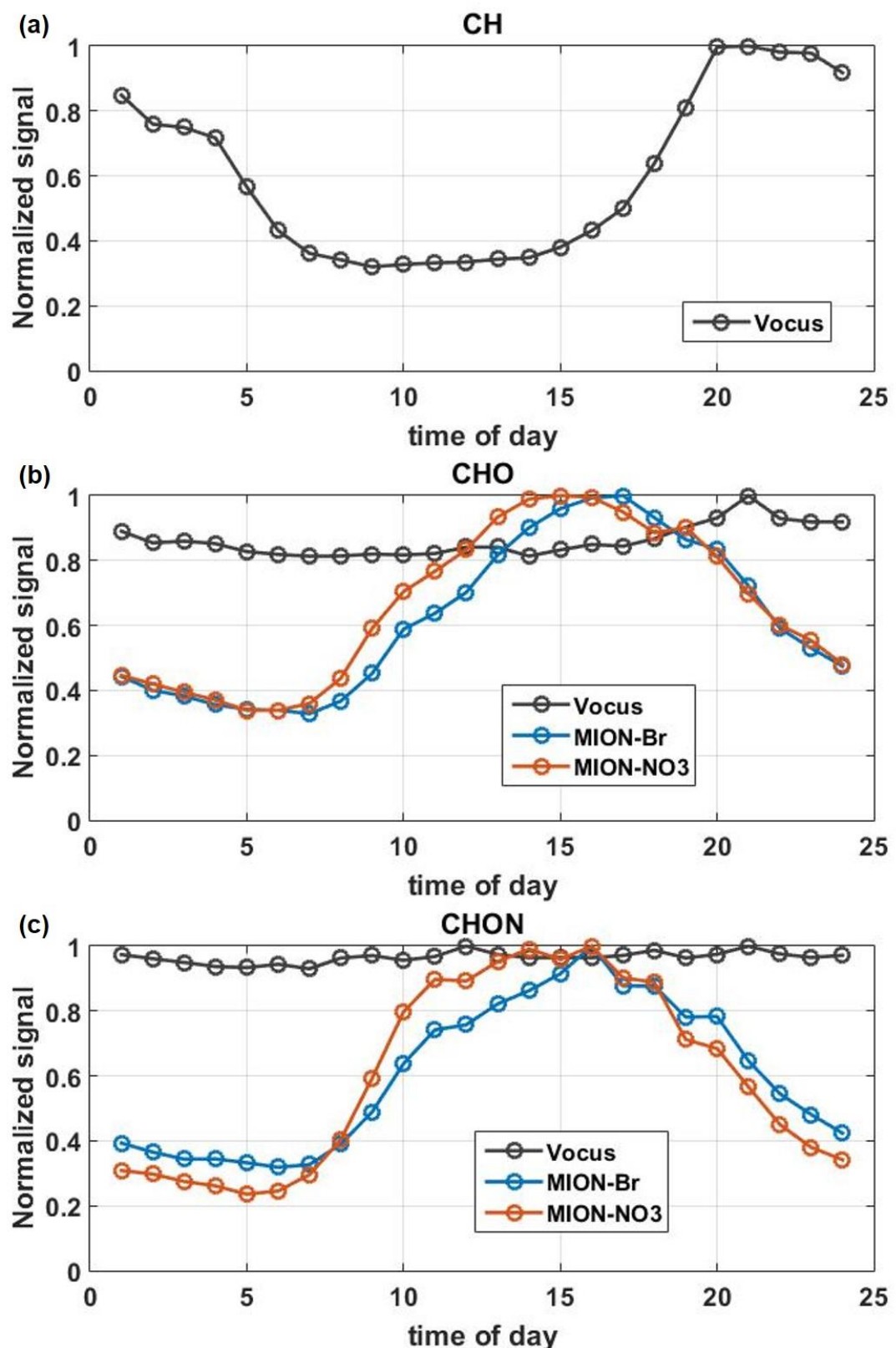

**Figure 4.** The median diurnal patterns of the total CH compounds measured by Vocus **(a)**, CHO **(b)**, and CHON compounds **(c)** measured by Vocus, MION-Br, and MION-NO₃ during the whole measurement period. Signals were normalized to their maximum values.

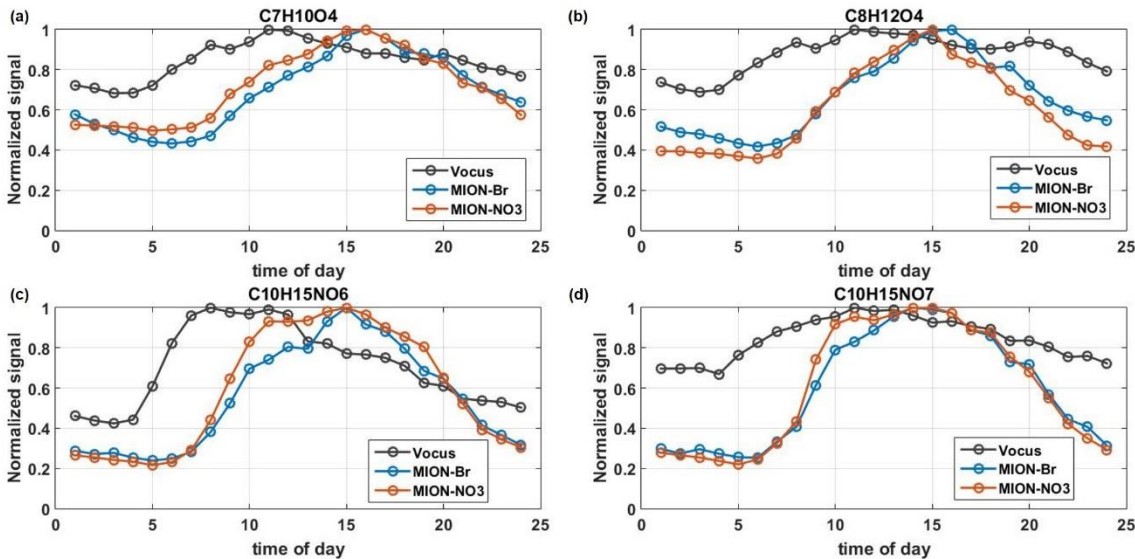

780

**Figure 5.** The median diurnal patterns of $C_7H_{10}O_4$ **(a)**, $C_8H_{12}O_4$ **(b)**, $C_{10}H_{15}NO_6$ **(c)**, and $C_{10}H_{15}NO_7$ **(d)** measured by Vocus, MION-Br, and MION-NO$_3$ during the whole measurement period. Signals were normalized to their maximum values.

**Table 2.** Contribution (%, average ± 1 standard deviation) of different compound groups to total organics measured by different measurement techniques based on the modified Li et al. (2016) approach (Daumit et al., 2013;Isaacman-VanWertz and Aumont, 2020).

| Compound group | Vocus | MION-Br | MION-NO$_3$ |
|---|---|---|---|
| ULVOC | / | 0.02 ± 0.04 % | 0.5 ± 0.6 % |
| ELVOC | / | 2.0 ± 1.8 % | 2.3 ± 1.7 % |
| LVOC | 0.02 ± 0.01 % | 5.6 ± 2.9 % | 11.6 ± 5.1 % |
| SVOC | 0.4 ± 0.2 % | 16.2 ± 4.9 % | 23.9 ± 5.1 % |
| IVOC | 45.8 ± 5.4 % | 65.8 ± 8.5 % | 56.3 ± 10.6 % |
| VOC | 53.7 ± 5.5 % | 10.4 ± 8.2 % | 5.4 ± 2.4 % |

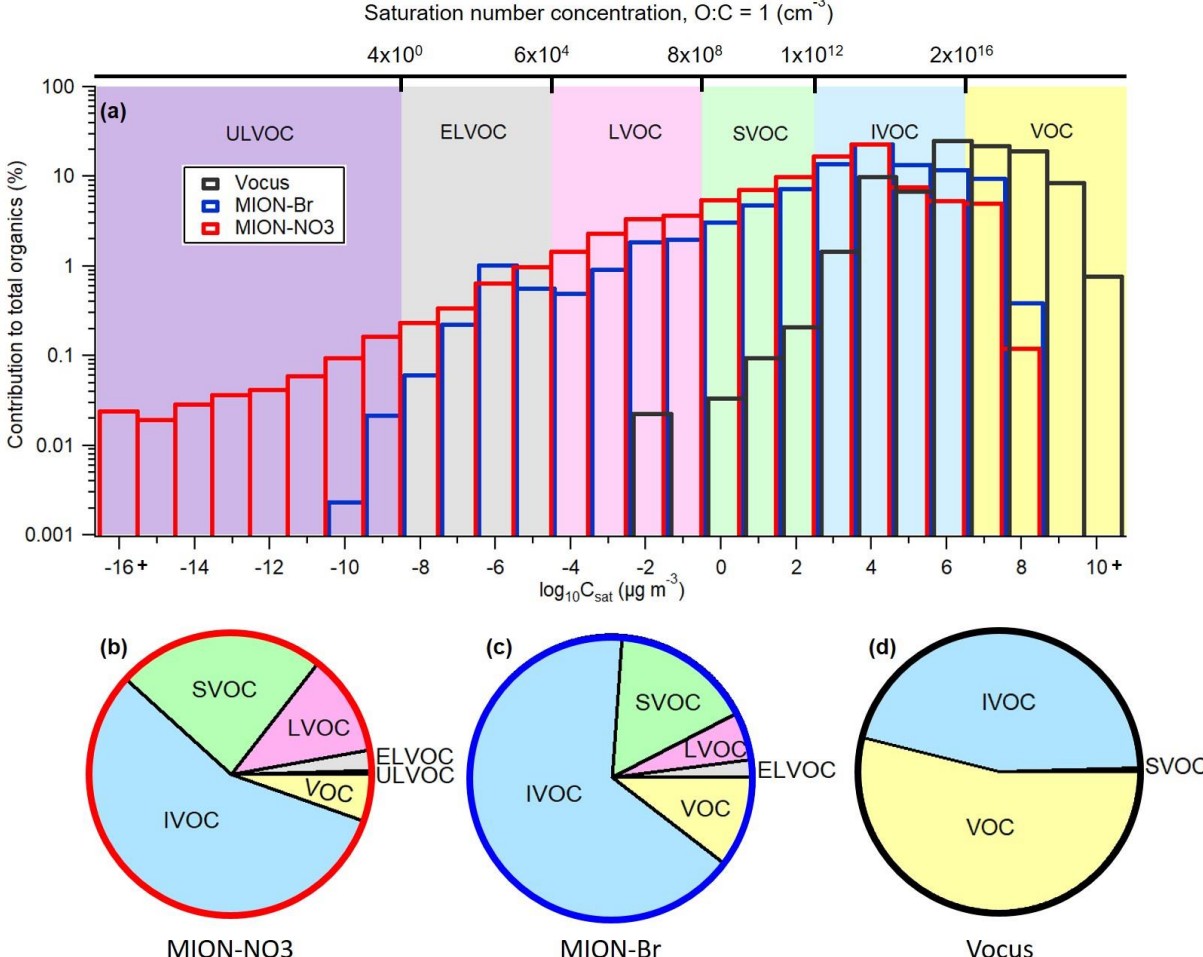

**Figure 6. (a)** Volatility distribution comparison for organic compounds detected by different measurement techniques and parameterized with the modified Li et al. (2016) approach (Daumit et al., 2013; Isaacman-VanWertz and Aumont, 2020); resulting pie charts for the contributions of VOC, IVOC, SVOC, LVOC, ELVOC, and ULVOC for MION-NO$_3$ **(b)**, MION-Br **(c)**, and Vocus **(d)**. Contribution of LVOC for Vocus (0.02 ± 0.01 %) and ULVOC for MION-Br (0.02 ± 0.04 %) were not labeled in the pie chart.

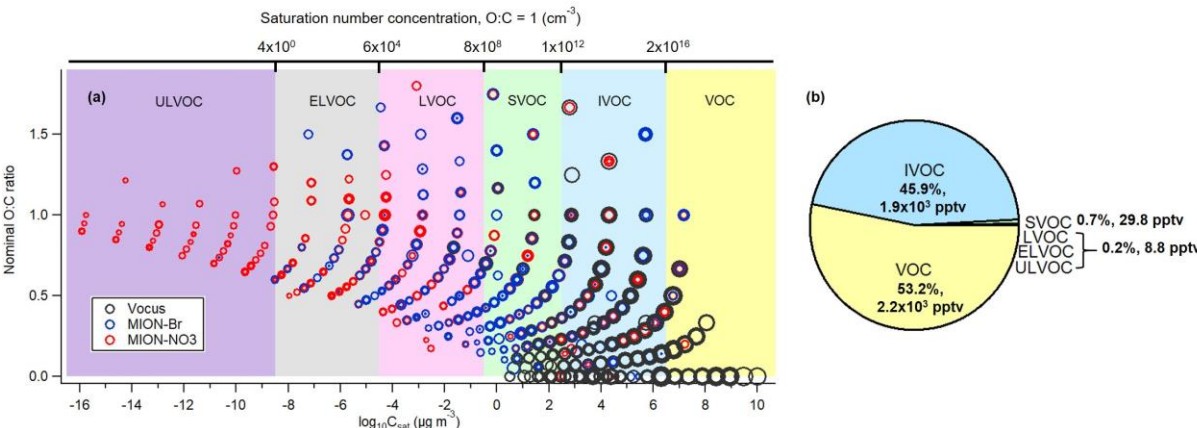

795    **Figure 7. (a)** Combined 2-dimentional volatility distribution for all measured organic compounds (with the approach described in section 2.2.1) parameterized with the modified Li et al. (2016) approach (Daumit et al., 2013; Isaacman-VanWertz and Aumont, 2020). Markers were sized by the logarithm of their corresponding concentrations, and marker color represents that either the compound was only measured by that instrument or the maximum concentration of the compound observed in common was detected by that instrument; **(b)** resulting pie

800    chart for the contributions of VOC, IVOC, SVOC, LVOC, ELVOC, and ULVOC.