# Peer review of "Figure S1. Overlaps of all organic compounds (a), CHO compounds (b), and CHON compounds (c) measured by Vocus, MION-Br, and MION-NO3."

_Atmospheric Chemistry and Physics, 2020_

## Referee Comment (RC1) · Anonymous Referee #1 · 2 Feb 2021

Comments on acp-2020-1257

In their manuscript "Measurement report: Molecular composition and volatility of gaseous organic compounds in a boreal forest: from volatile organic compounds to highly oxygenated organic molecules", the authors present data from summer in a boreal forest using a suite of state-of-the-art mass spectrometers. Overall, it is scientifically valid work, and advances the important work of trying to understand the different strengths and limitations of the many new CIMS approaches. My primary concern with this work is that it is not wholly clear to me that it should be published in ACP instead

of AMT. While the title pitches the science of the work, it is clear from the figures and most of the discussion that the bulk of this work is on intercomparisons between the 3 instruments(/instrument modes) and how they complement each other. While there are some plots of distributions and diurnals, etc., the focus of the discussion of these figures is again on the instrumentation. Overall, the bulk of the science here (and in my opinion, a lot of the highest value) is in the "Measurement Techniques" not the "Chemistry and Physics". In that context, I do have some technical concerns noted in my general comments below, but these are mostly addressable through changes in language and discussion and minor re-processing. I think this can and should be published with relatively minor revisions, but I'm not totally convinced that should happen in this specific journal.

General comments

(1) There are some minor english issues - nothing egregious but quite a few cases of odd sentence structures. One of the english-as-a-first-language authors on this work should copy edit more closely.

(2) In parameterizing Csat, Li et al. has a problem with nitrogen. The empirical approach was derived with very few nitrate groups, so treats nitrogen essentially as an amine. In environments where NO3 is expected to be a dominant form of organic nitrogen, this can bias the vapor pressure low by roughly two orders of magnitude per nitrogen atom. Probably not enough to change any conclusions, but with CHON representing roughly a quarter of the MION mass, it is probably enough to shift some distributions around a little. A recent paper in review in this journal describes this issue (https://acp.copernicus.org/preprints/acp-2020-1038/acp-2020-1038.pdf) , and proposes a solution by treating NO3 units a OH groups, following the approach used by Daumit et al. in their parameterization.

(3) While this paper focuses most heavily on intercomparisons between the instruments, considerations of some of the pitfalls of these tools are not discussed. For

example, though the potential presence of isomers is discussed in a few places, it tends to be glossed over based on relatively weak assumptions (e.g., different isomers probably have different diurnals). While a few specific spots are described below, I would just more generally caution the authors to consider that it is quite likely that the presence of isomers is the rule, not the exception, at that different isomers may have significantly different instrument sensitivities, the the authors should keep this in mind as they interpret their results. Its not really clear to me why diurnals tend to be the metrics by which isomer composition is being compared - why not point-by-point correlations, which should be high if they are truly the same isomers? One suggestion is, while isoprene is lower than monoterpenes, you may see the C5-methyltetrols (C5H12O4). This specific species is helpful because there are not a lot of likely ways to draw that formula since it is saturated and a dominant isoprene product (though there are a few peroxide options), so if multiple intruments see it, it might give some benchmark as to how correlated ions might be when they are very likely the same set of isomers.

Similarly, the role of fragmentation in these results is not considered deeply, though PTR is known to fragment. What does this mean for measured mass? For instrument overlap? For average elemental composition? Is this related to the flat diurnal in Vocus CHO and CHON? These instruments are amazing advances, but they do have limitations in their interpretation, and these limitations are not always deeply considered in this work.

Technical comments "VOC" is usually pluralized as VOCs when used in a plural sense.

Line 70-71. This sentence isn't quite grammatically correct, re-word.

Line 73-76. Run-on sentence, somewhat confusing.

Line 74. Why isn't Interface capitalized in APi-ToF?

Line 75. It's not clear to me: is MION just a switching reagent ionization approach,

which has been shown previously using a PTR, but as applied to an API-ToF? Because it is discussed in the same "breath" as the the Vocus, my initial reading is that it is a functionality of the Vocus, but I gather that the MION instrument is a physically distinct ToF-CIMS. This confusion makes it a bit hard to understand or parse the rest of this paragraph. I think this paragraph just needs some editing and further clarification and detail.

Line 108. "Finnish winter time" is confusing, since it is in spring. Perhaps "Finnish Winter Time" or "Eastern European Time (EET)" which I think is the general term for UTC+2 in Europe.

Line 126-129. Is the assumption the sulphuric acid represents the kinetic limit sensitivity, so is used as a floor for org? If so, that should be explicitly mentioned, otherwise it's not clear why the org sensitivity factors are being determined by sulphuric acid.

Line 134-137. Similar diurnal patterns is a poor approach to determining isomer content. Take, for example, monoterpenes, for which there are usually around a dozen isomers, but all are expected to have a similar diurnal. Point-to-point correlations (R2) might be a better metric than diurnals, but it will still suffer from this example issue (just perhaps less so). Since you are comparing across two different ionizations, this approach is perhaps a bit more reasonable (if an ionization scheme sees one group of isomers, the other one probably does too), but it is still has serious issues. Isomers can vary in their senstivity by an order of magnitude within an ionization scheme (e.g., iodide, Lee et al. CITE), so one ionization scheme could see one set of isomers with high sensitivity, and the other could see a different set with high sensitivity, but these could still have similar diurnals. All-in-all, I'm sympathetic to the need to do something about potential overlap and the uncertainties in bulk calibration of CIMS, but scaling one instrument to another based on diurnals is built on fairly shaky assumptions that need a more robust examination. Are there trends in correlations between ionizations as a funciton of ion elemental ratios that might allow you to tease out when they are seeing the same isomers and when they are not? Or any other features within the

data? Simply put, similar diurnals is insufficient evidence for "likely to be the same species", and more caution is warranted in acting on this conclusion.

Line 155-157. This is a better/more conservative approach to handling overlap.

Line 182. A Pt100 should be defined/described

Line 205. Is this the full list of compounds excluded, or just an example list? Is the full list provided somewhere? Would it be helpful to also add that data, for instance as Vocus_LL or some other signifier? That would be interested from an organic carbon budget perspective.

Line 233. Is this average composition of organic gases, or does it include ACSM-measured organic particles?

Line 238. "followed-by groups" is not something I've seen before in written English

Lines 226-243. While PTR is fairly soft, it is known to have non-negligible fragmentation (Yuan et al., 2017, e.g., Figure 5 therein). How might this impact both the quantification of the total measurement by this instrument, and/or understanding of the elemental compositions? This issue of course does not involaidate the Vocus, or these measure-ments, but the effects of fragmentation and its impacts on the potential interpretation and conclusions in this work should be considered and discussed.

Line 268. It's not clear to me the C20 is necessarily diterpenes. While the SI does show some diterpenes (which is very exciting and interesting, and sadly buried in the SI), monoterpenes are known to dimerize and for C20 compounds. I note that the bar on C20 looks like it is mostly O>=12, so highly oxygenated. Is this not just monoterpene dimers? It might provide some insight into the influence of monoterpene-dimers vs. diterpene-monomers to looks at distributions of oxygen number.

Line 284-286. Why would this smearing occur for Vocus data, but not the other data? Is it related to the tendency for nitrates to fragment in PTR?

FIgure 1. Why use ppb for inorganics and cm-3 for organics? Organic gases are more commonly reported as ppb.

Figures 6 and 7. I recognize why the authors chose to plot these distributions on a log scale, but a bar chart on a log scale is inherently inaccurate/confusing, especially a stacked bar chart. Because there is no "zero", drawing a line to zero on a bar chart creates a wholly arbitrary scaling, which means the bar size is no longer in any way proportional to quantity. Consider Figure 7 at log(Csat)=7. While roughly 99.9% of the concentrationis measured by the Vocus, more than half the bar is blue. At the same time, if the bars were stacked in the opposite order, MION-Br would be negligibly small sitting on top of the Vocus. Similarly, the scale on the x-axis could reasonably be altered to start at 10^3 or 10^2 instead of 10^4, and that choice would dramatically change the areas of only the bottom bar in the stack. What the solution here is, I'm not sure, but I strongly recommend the authors make some other style choice.

Is it worth splitting these figures across two figures? It seems to me that 6a and 7a are showing basically the same data - couldn't you should had 6b-d to Figure 7. Relatedly, though they seem to be plotting the same data, I can't reconcile them quantitatively. Again, as an example log(Csat)=7. In Figure 6a, this looks like roughly: Vocus 15%, Br 8%, NO3 3%. In Figure 7a, the ratio is Vocus: stack from 10^7.5 to 10^10 = 10^10, Br: stack to 10^7.5, NO3:negligible. That is a ratio of Vocus:Br = 300:1 instead of 2:1. I wonder of this issue is related to the stacked log plot issue described above.

---

## Referee Comment (RC2) · Anonymous Referee #2 · 4 Mar 2021

Huang and coauthors compared the measurements of gas-phase organic compounds by Vocus PTR-ToF, Nitrate CIMS and a Br CIMS. They found different chemical compositions from the three different techniques. The measured diurnal profiles from the three techniques are different even for compounds with the same molecular. The authors claimed that a more comprehensive understanding of molecular composition and volatility can be obtained by this kind of comparison and combined analysis. This manuscript is generally well written. I can be accepted in Atmospheric Chemistry and Physics, after addressing my following comments.

[Figure]

(1) Line 115, a stainless-steel tube of 0.9 m long inlet was used for the MION API-TOF. Will SVOC and HOMs loss to the stainless-steel tube. Why not using PFA, See Deming et al., 2019 AMT.

(2) Line 120: I am not sure about how data processing was done for Br CIMS. As Bromine has two isotopes, 79 and 81. Then, each compound would generate at least two product ions, even there is no fragmentation or other chemical pathways. Did the author take into account both, or just one? Will this cause problem to detect compounds with two hydrogen apart (e.g. CxHyOz and CxHy+2Oz)?

(3) Line 120-125: As Br CIMS is kind of new reagent ion, can the authors provide some information about the types of compounds can be measured by Br CIMS. It would be if the advantages and also disadvantages for Br CIMS can be provided somewhere in the manuscript.

(4)Line 136: why to scale the measurement of Br CIMS, how 0.3 is obtained. Are you claiming the sensitivity variations are same between NO3- and Br-. As many of the conclusions rely on good quantification for all of the instruments, a better of quantification of Br CIMS should be conducted.

(5)Line 150: The quantification of PTR-TOF is also way too simple. It would be better to use the relationship between the kinetic reaction rate constants (H3O+ with VOCs) and calibrated sensitivity (Sekimoto et al., 2017 IJMS; Yuan et al., 2017 CR).

(6) Line 160-175: Could the authors comment on the uncertainties form the calculation of volatility from the parameterization method.

(7) Figure 1: why Br CIMS has more data missing than NO3- CIMS, for example the period around May 17, as this is achieved by the same instrument.

(8) Line 275-280: can the authors also provide the comparison of time series for some of the important ions. May be also their correlation. It is expected PTR-TOF would measure more species, as almost all OVOCs has signals in the mass spectra with

similar sensitivities. It might be due some of the isomers are not measured by Br CIMS and NO3- CIMS.

---

## Author Comment (AC1) · 14 Apr 2021

**Responses to reviewers' comments for manuscript**

Measurement report: Molecular composition and volatility of gaseous organic compounds in a boreal forest: from volatile organic compounds to highly oxygenated organic molecules

Wei Huang1,†,\*, Haiyan Li2,†, Nina Sarnela1, Liine Heikkinen1, Yee Jun Tham1, Jyri Mikkilä3, Steven J. Thomas1, Neil M. Donahue4, Markku Kulmala1, and Federico Bianchi1,\*

1Institute for Atmospheric and Earth System Research / Physics, Faculty of Science, University of Helsinki, Helsinki, 00014, Finland

2School of Civil and Environmental Engineering, Harbin Institute of Technology, Shenzhen, 518055, China

3Karsa Oy., A. I. Virtasen aukio 1, Helsinki, 00560, Finland

4Center for Atmospheric Particle Studies, Carnegie Mellon University, 5000 Forbes Avenue, Pittsburgh, PA 15213, USA

†*These authors contributed equally to this work.*

\*Correspondence to: Wei Huang (wei.huang@helsinki.fi) and Federico Bianchi (federico.bianchi@helsinki.fi)

We thank all the reviewers for their evaluation of the manuscript, and for their constructive feedback. Replies to the individual comments are directly added below in italics in green, and changes in the manuscript in italics in blue. Please note that only references that are part of the replies to the comments are listed in the bibliography at the end of this document. References in copied text excerpts from the manuscript are not included in the bibliography. Page and line numbers refer to the original manuscript text.

**Reviewer #1** (responses in italics)**

In their manuscript "Measurement report: Molecular composition and volatility of gaseous organic compounds in a boreal forest: from volatile organic compounds to highly oxygenated organic molecules", the authors present data from summer in a boreal forest using a suite of state-of-the-art mass spectrometers. Overall, it is scientifically valid work, and advances the important work of trying to understand the different strengths and limitations of the many new CIMS approaches. My primary concern with this work is that it is not wholly clear to me that it should be published in ACP instead of AMT. While the title pitches the science of the work, it is clear from the figures and most of the discussion that the bulk of this work is on intercomparisons between the 3 instruments(/instrument modes) and how they complement each other. While there are some plots of distributions and diurnals, etc., the focus of the discussion of these figures is again on the instrumentation. Overall, the bulk of the science here (and in my opinion, a lot of the highest value) is in the "Measurement Techniques" not the "Chemistry and Physics". In that context, I do have some technical concerns noted in my general comments below, but these are mostly addressable through changes in language and discussion and minor re-processing. I think this can and should be published with relatively minor revisions, but I'm not totally convinced that should happen in this specific journal.

We thank the reviewer for his/her overall comments on this manuscript. We understand the reviewer's concerns and we can imagine that using this suite of instruments could have led the readers to that conclusions. However, we would like to stress here, that the comparison is a fundamental part in order to understand the scientific results presented in this study. In fact, we do show some instrumental comparison results in the manuscript, but these are needed to fulfill the aim of this work, i.e., to provide a complete picture and full understanding of the molecular composition and volatility of ambient gaseous organic compounds (covering from volatile organic compounds, VOCs, to highly oxygenated organic molecules, HOMs) in a chemically rich environment. To our knowledge, this has never been achieved so far. In order to achieve this, we therefore needed to deploy and combine different measurement techniques (e.g., Vocus, MION-Br, and MION-NO3). Without the complementary techniques, one might arrive at one-sided understandings of the "real" picture of the gaseous organic compounds in the field. In our manuscript, the complementary information obtained from Vocus, MION-Br, and MION-NO3 provides the bulk molecular composition, oxidation extent, as well as volatility of all gaseous organic compounds (from VOCs to HOMs) in this boreal forest (see also the 2-dimentional volatility distribution in Figure R1). These results therefore provide a more comprehensive understanding of the molecular composition and volatility of atmospheric organic compounds, as well as a better basis to test and improve parameterizations for predicting organic compound evolutions in transport and climate models. We therefore believe that this study is not to introduce new "Measurement Techniques" or compare the performance of different instruments, but to target the scientific question we would like to address in the manuscript. We understand the reviewer concerns and in order to make our message clear also to future readers we have emphasized this, by adding this 2-dimentional volatility distribution figure as Figure 7a as well as the following sentences/information to the manuscript. The original Figure 7a-b are now Figure 7b-c.

Line 15 (Abstract): "In order to obtain a complete picture and full understanding of the molecular composition and volatility of ambient gaseous organic compounds (from volatile organic compounds, VOCs, to highly oxygenated organic molecules, HOMs), two different instruments were used. A Vocus [...]".

Line 85-86 (Section 1, 2nd paragraph): "[...] it still remains challenging to provide a comprehensive understanding of the molecular composition and volatility of both VOCs and OVOCs, particularly in the field. And to our knowledge this has never been achieved so far."

Line 331-332 (Section 3.4,  $3^{rd}$  paragraph): "[...] a combined volatility distribution was plotted to obtain the bulk volatility of all measured organic compounds (with the approach described in section 2.2.1) at our measurement site (Figure 7). The combined volatility distribution covers very well from VOCs to HOMs, with varying O:C ratios and volatility ranges (Figure 7a). It therefore provides a more complete picture of the volatility distribution of gaseous organic compounds in this boreal forest. The average mass-weighted  $log_{10}C_{sat}$  value representing the bulk of all measured gaseous organic compounds in this boreal forest was ~6.1 µg m-3. In general [...]".

Line 348-349 (Section 4, 1st paragraph): "In this paper, with an aim of obtaining a complete picture from VOCs to HOMs, the molecular composition and volatility of

gaseous organic compounds were investigated with the deployment of a Vocus and a MION API-ToF [...]".

Line 356 (Section 4,  $1^{st}$  paragraph): "The average mass-weighted chemical composition representing the bulk of all measured gaseous organic compounds in this boreal forest was  $C_{6.0}H_{8.7}O_{1.2}N_{0.1}$ , indicative of the short carbon backbone and relatively low oxidation extent. Besides, [...]".

Line 362 (Section 4,  $2^{nd}$  paragraph): "The average mass-weighted  $log_{10}C_{sat}$  value representing the bulk of all measured gaseous organic compounds in this boreal forest was ~6.1 µg m-3. In addition, the VOC [...]".

**Figure R1.** Combined 2-dimentional volatility distribution for measured organic compounds parameterized with the modified Li et al. (2016) approach (Daumit et al., 2013;Isaacman-VanWertz and Aumont, 2020). Markers were sized by the logarithm of their corresponding concentrations, and marker color represents that either the compound was only measured by that instrument or the maximum concentration of the compound observed in common was detected by that instrument.

**General comments:**

1. There are some minor english issues - nothing egregious but quite a few cases of odd sentence structures. One of the english-as-a-first-language authors on this work should copy edit more closely.

The odd sentence structures listed by the reviewer in "Technical comments" were re-

**worded according to the reviewer's suggestion.**

2. In parameterizing Csat, Li et al. has a problem with nitrogen. The empirical approach was derived with very few nitrate groups, so treats nitrogen essentially as an amine. In environments where NO3 is expected to be a dominant form of organic nitrogen, this can bias the vapor pressure low by roughly two orders of magnitude per nitrogen atom. Probably not enough to change any conclusions, but with CHON representing roughly a quarter of the MION mass, it is probably enough to shift some distributions around a little. A recent paper in review in this journal describes this issue (https://acp.copernicus.org/preprints/acp-2020-1038/acp-2020-1038.pdf), and proposes a solution by treating NO3 units a OH groups, following the approach used by Daumit et al. in their parameterization.

That is a very good point and we thank the reviewer for bringing this up. We have now recalculated the VBS distributions by treating NO3 units as OH groups following the approach by Daumit et al. (2013) and Isaacman-VanWertz and Aumont (2020). The resulting VBS distribution and compound group contribution changed a little and didn't influence our conclusions. We have updated the Figures and Table (Figure 6, 7, S10, S11, and Table 2) and the corresponding numbers in the texts. The updated VBS parameterization information was also added/rephrased in the manuscript as following:

Lines 173 (Section 2.2.2,  $1^{st}$  paragraph): "[...] these "b" values can be found in Li et al. (2016). Due to that the empirical approach by Li et al. (2016) was derived with very few organonitrates and could therefore lead to bias for the estimated vapor pressure (Isaacman-VanWertz and Aumont, 2020), we modified the  $C_{sat}$  (298 K) of CHON compounds by replacing all NO3 groups as OH groups (Daumit et al., 2013)."

Line 305-306 (Section 3.4,  $1^{st}$  paragraph): "Based on the  $log_{10}C_{sat}$  values of all organic compounds parameterized with the modified Li et al. (2016) approach (Daumit et al., 2013; Isaacman-VanWertz and Aumont, 2020) described in section 2.2.2, they [...]".

3. While this paper focuses most heavily on intercomparisons between the instruments, considerations of some of the pitfalls of these tools are not discussed. For example, though the potential presence of isomers is discussed in a few places, it tends to be glossed over based on relatively weak assumptions (e.g., different isomers probably have different diurnals). While a few specific spots are described below, I would just more generally caution the authors to consider that it is quite likely that the presence of

isomers is the rule, not the exception, at that different isomers may have significantly different instrument sensitivities, the authors should keep this in mind as they interpret their results. Its not really clear to me why diurnals tend to be the metrics by which isomer composition is being compared - why not point-by-point correlations, which should be high if they are truly the same isomers? One suggestion is, while isoprene is lower than monoterpenes, you may see the C5-methyltetrols (C5H12O4). This specific species is helpful because there are not a lot of likely ways to draw that formula since it is saturated and a dominant isoprene product (though there are a few peroxide options), so if multiple instruments see it, it might give some benchmark as to how correlated ions might be when they are very likely the same set of isomers.

We agree with the reviewer that different instruments may have very different sensitivities towards isomers. As also pointed out by the reviewer in the "Specific comments" 8, even with the same diurnal patterns and high point-by-point correlations, it's still possible to be different isomers (e.g., different isomers of monoterpenes could have similar diurnal patterns).

Therefore, we have abandoned the scaling approach through comparing the diurnal patterns of organic compounds observed in common by MION-Br and MION-NO3. Instead, after comparing the ambient sulphuric acid concentrations measured by *MION-Br* and *MION-NO*3 (See Figure R2a), we scaled the sulphuric acid calibration factor of MION-Br to that of MION-NO3. The reason why we scaled the sulphuric acid calibration factor of MION-Br to that of MION-NO3 is because Br mode has been found to be more sensitive to RH (Hyttinen et al., 2018) and the high RH in the calibration kit (Kürten et al., 2012) could cause some uncertainties in its calibration factor. This scaling approach is more reasonable since the calibrations were done for sulphuric acid (compound representing the kinetic limit sensitivity; Viggiano et al., 1997; Berresheim et al., 2000) for MION-Br and MION-NO3. The scaling factor of sulphuric acid was determined to be 0.53 (median value; see Figure R2b). We have therefore deleted the sentence in Line 133-137 and added the following sentence in Line 130 (Section 2.2.1, 1st paragraph) of the manuscript: "By comparing the ambient  $H_2SO_4$  concentrations measured by MION-Br and MION-NO3, the median value (0.53) was used to scale down the  $H_2SO_4$  concentration measured by MION-Br, due to that the high RH in the calibration kit could cause some uncertainties in its calibration factor (Hyttinen et al., 2018; Kürten et al., 2012)." The corrected organic

concentrations for MION-Br were also updated in Figure 1, 2, 7, S5, S10, and S11.

Besides, based on the reviewer 1's and reviewer 2's suggestion we have also calculated the correlation coefficients for several dominant CHO and CHON species (including  $C_7H_{10}O_4$ ,  $C_8H_{12}O_4$ ,  $C_{10}H_{15}NO_{6-7}$ ) discussed in the manuscript, for a simplified examination of isomer content for individual compound (see Table R1). The corresponding information was added to Line 293 (Section 3.3, 2nd paragraph) of the manuscript: "[...] The inconsistent trends in time series and the varying correlations of these above-mentioned dominant CHO and CHON species indicate different isomer contributions detected by different measurement techniques (Figure S8 and Table S3). Similar behaviors were also evident for [...]". The original Figure S7-S9 are now Figure S9-S11.

---

## Author Response (AR2)

**Responses to Editor's comments for manuscript**

Measurement report: Molecular composition and volatility of gaseous organic compounds in a boreal forest: from volatile organic compounds to highly oxygenated organic molecules

Wei Huang1,†,\*, Haiyan Li2,†, Nina Sarnela1, Liine Heikkinen1, Yee Jun Tham1, Jyri Mikkilä3, Steven J. Thomas1, Neil M. Donahue4, Markku Kulmala1, and Federico Bianchi1,\*

1Institute for Atmospheric and Earth System Research / Physics, Faculty of Science, University of Helsinki, Helsinki, 00014, Finland

2School of Civil and Environmental Engineering, Harbin Institute of Technology, Shenzhen, 518055, China

3Karsa Oy., A. I. Virtasen aukio 1, Helsinki, 00560, Finland

4Center for Atmospheric Particle Studies, Carnegie Mellon University, 5000 Forbes Avenue, Pittsburgh, PA 15213, USA

†*These authors contributed equally to this work.*

\**Correspondence* to: Wei Huang (wei.huang@helsinki.fi) and Federico Bianchi (federico.bianchi@helsinki.fi)

**Comments to the Author:**

Dear Authors:

Thank you for your consideration of the referee comments. I think that the comments have largely been adequately addressed and I am happy to accept the paper subject to consideration of the minor comments below. Line numbers refer to the track changes version of the manuscript.

We thank the Editor for the positive and constructive feedback of our revision. Replies to the individual comments are directly added below in italics in green, and changes in the manuscript in italics in blue. Please note that only references that are part of the replies to the comments are listed in the bibliography at the end of this document. References in copied text excerpts from the manuscript are not included in the bibliography. Line numbers refer to the updated track changes version of the manuscript.

1) I agree with referee 1 that stacked bar charts using a log scale are confusing/can easily lead to misinterpretation. Here I am specifically referring to Figure 7b and Figures S11. In my opinion, the non-stacked plots of Fig 6 are easier to interpret. In a stacked bar chart, one wants to connect the size of the bar to the relative importance - something that is not easily done on a log scale. Personally, I am unsure what additional information Figure 7b adds that isn't captured by Fig. 7c and S10. In my opinion, Figure S10 does a better job of representing the relative contribution of each instrument. If more information about total concentrations wanted to be conveyed, one could consider sizing the pie charts to the total

concentration. I urge the authors to consider these points and to think about improving the representation of the data provided in Figures 7b and S11.

We agree with the Editor that the information we want to convey has already been captured by Figure 7c and S10. Therefore we deleted Figure 7b, S10a, and S10c, as well as the corresponding sentence in Line 386; and we added the total concentrations and relative contributions for each volatility group to the corresponding pie charts. The original Figure 7c, S10b, and S10d are now Figure 7b, S10a, and S10b, respectively.

2) I suggest clarifying the first sentence of the captions for Figures 6 and 7 to more clearly articulate how these figures show different data. The main text does this well, but the legends would be unclear to anyone skimming the figures only.

The first sentence of the caption for Figure 6 was changed to "Volatility distribution comparison for organic compounds detected by different measurement techniques and parameterized with the modified Li et al. (2016) approach (Daumit et al., 2013;Isaacman-VanWertz and Aumont, 2020)."; and the first sentence of the caption for Figure 7 caption was changed to "Combined 2-dimentional volatility distribution for all measured organic compounds (with the approach described in section 2.2.1) parameterized with the modified Li et al. (2016) approach (Daumit et al., 2013;Isaacman-VanWertz and Aumont, 2020)." In addition, in order to make it even more obvious that the pie charts described different instruments in Figure 6, we added stroke colors on the edge of the pie charts in Figure 6b-d which are corresponding to the colors of the histogram in Figure 6a, and reordered the pie charts so that they scan the same way as the VBS (Figure 6a).

3) Line 16: I suggest "In order to obtain a more complete ..." as there may be compounds not measured by any of these techniques (or compounds lost to inlets, etc.).

The Editor is right. The sentence was changed.

4) Line 94 "And to our knowledge ...." I suggest removing this added sentence as it detracts from the paper by overselling the work.

**Sentence removed.**

5) Line 122: Please explain the zeroing mode for the MION in slightly more detail. Zeroing for which configuration?

Zeroing mode was done by removing all natural ions with an ion-filter. The information for zeroing mode was added to the manuscript as following: "[...] followed by 2 min of ion-filter zeroing for the API mode before switching from API mode to the next mode."

6) Line 147): A few more details regarding how the high RH can cause uncertainties is warranted given how much of the manuscript focuses on quantification. Is the known bias consistent with the factor of 2 that the MION-Br is scaled by? How role do changes in ambient RH play?

We understand the Editor's concern about humidity effect on instrumental sensitivity. Due to that the sulphuric acid calibration method was originally developed for NO3-CIMS (Kürten et al., 2012), potential interfering processes caused by the high water vapor concentrations inside the calibration kit  $(\sim 5 \times 10^{14-16} \text{ cm}^{-3})$  might exist for the Br-CIMS case and therefore might lead to some uncertainties in its sulphuric acid calibration factor (Kürten et al., 2012; Hyttinen et al., 2018). This is also shown in the inconsistent time series of sulphuric acid concentrations measured by MION-Br and MION-NO3. Although the humidity dependence (i.e., water binding strength) using  $Br^-$  as the reagent ion is stronger than that using  $NO_3^-$  as the reagent ion (Hyttinen et al., 2018), the Br-CIMS sensitivity has been found to be invariant with RH higher than 10% for e.g. hydroperoxyl radicals (Sanchez et al., 2016), which is the case during our measurement period (RH between 20–100 %; see Figure R1). Despite of the large variability in ambient RH (Figure R1), the highly consistent and highly correlated time series of the concentrations of e.g. several dominant CHO and CHON species (including C7H10O4, C8H12O4,  $C_{10}H_{15}NO_{6-7}$ ) measured by MION-Br with those measured by MION-NO3 (see Figure S8 and Table S3) support this statement, indicating insignificant effect of RH on the OVOC measured by MION-Br at our measurement site. In addition, water clustered with Br- has also been included in the signal normalization to account for the humidity effect on reagent ion competition, both for the calibration data and ambient data (see section 2.2.1, 1st paragraph). For better clarification, this sentence was rephrased as following: "[...] due to that the high water vapor concentrations in the calibration kit (~ $5 \times 10^{14-16}$  $cm^{-3}$ ) might cause some uncertainties in the H2SO4 calibration factor of MION-Br (Hyttinen et al., 2018; Kürten et al., 2012). However, the MION-Br sensitivity has been found to be invariant with the measured ambient RH at our measurement site (20–100 %) for e.g. hydroperoxyl radicals (Sanchez et al., 2016), and the water clustered with  $Br^-$  has also been included in the signal normalization of organic compounds to account for the humidity effect on reagent ion competition (see equation (1))."

Figure R1. Time series of relative humidity (RH) during the measurement period.

**but it would be a better fit elsewhere. Perhaps in section 2.2.2.**

The information was moved to the last sentence of section 2.2.2: "Besides, the fragmentation of organic compounds inside the instruments (e.g., Vocus) may also bias the  $C_{sat}$  results towards higher volatilities, resulted from the signal bias of parent ions towards lower values and of fragment ions towards higher values (Heinritzi et al., 2016)."

8) Line 350-352: Based on your measurements, you know the answer to the first possibility listed (different saturation levels) whereas the other two are more difficult to assess with the current instruments. Therefore, it isn't really appropriate to list the three together. And the first one should probably be investigated more quantitatively.

For compounds with the same number of carbon and oxygen atoms but different hydrogen atoms (i.e., different saturation level), we chose C10H2O3 and C10H2NO4 as examples to compare their behaviors. As shown in Figure R2, compounds with different saturation level varied, possibly due to different chemistry involved in their formation (Zhao et al., 2018; Molteni et al., 2019). Even compounds with the same molecular formula were found to behave differently among different measurement techniques (see also Figure 5), probably resulted from different parent compounds inside the instruments (e.g., Heinritzi et al., 2016; Zhang et al., 2017). Figure R2 was added to Figure S9, and the sentence was added/rephrased as following: "Compounds with the same number of carbon and oxygen atoms but different hydrogen atoms (i.e., different saturation level) were also found to behave differently (see Fig. S9c–d), possibly due to different chemistry involved in their formation (Zhao et al., 2018; Molteni et al., 2019). Even compounds with the same number of carbon and oxygen atoms but different hydrogen atoms (i.e., different saturation level) were also found to behave differently (see Fig. S9c–d), possibly due to different chemistry involved in their formation (Zhao et al., 2018; Molteni et al., 2019). Even compounds with the same molecular formula varied among different measurement techniques (see Fig. S9c–d and also Fig. 5). The differences can be likely resulted from different isomers detected by the different techniques, and/or fragmentation products from different parent compounds inside the instruments (e.g., Heinritzi et al., 2016; Zhang et al., 2017)."